# Restoring drifted electron microscope volumes using synaptic vesicles at sub-pixel accuracy

Hans Jacob Teglbjærg Stephensen [1], Sune Darkner[1] & Jon Sporring [1✉]

Imaging ultrastructures in cells using Focused Ion Beam Scanning Electron Microscope (FIB-SEM) yields section-by-section images at nano-resolution. Unfortunately, we observe that FIB-SEM often introduces sub-pixel drifts between sections, in the order of 2.5 nm. The accumulation of these drifts significantly skews distance measures and geometric structures, which standard image registration techniques fail to correct. We demonstrate that registration techniques based on mutual information and sum-of-squared-distances significantly underestimate the drift since they are agnostic to image content. For neuronal data at nano-resolution, we discovered that vesicles serve as a statistically simple geometric structure, making them well-suited for estimating the drift with sub-pixel accuracy. Here, we develop a statistical model of vesicle shapes for drift correction, demonstrate its superiority, and provide a self-contained freely available application for estimating and correcting drifted datasets with vesicles.

[1] Department of Computer Science, University of Copenhagen, Copenhagen, Denmark. ✉email: sporring@di.ku.dk

In three-dimensional (3D) scanning methods, such as focused ion beam scanning electron microscope (FIB-SEM), serial block-face imaging (SBF-SEM) and serial section transmission electron microscopy (SS-TEM), the scanning method alternates between forming a two-dimensional (2D) image and removing a layer of material. For 3D geometrical analysis, the sequence of 2D images must be recombined into a single, 3D image, which, unfortunately, is non-trivial. Sectioning of the tissue, as well as the subsequent imaging by electrons, often introduces a sideways sub-pixel translation between sections known as drift. In FIB-SEM, the drift may arise from a variety of practically uncontrollable factors, such as bending of the electron beam due to a charge gradient in the material and physical movement of the entire sample. Uncorrected drifts skew 3D distances, thus introducing errors in subsequent statistical and geometric 3D analyses, and in turn, on possible biological conclusions. Figure 1 shows an example of an ultrastructure brain region from a healthy adult rodent with easily noticeable drift when the dataset is viewed across multiple image planes. Datasets such as this have been and are still used actively[1–3] with no apparent mention of the correction for potential drift. In these works, it is unclear what effect such misalignment may have had on the presented results. In other works[4–8], the correction has been performed using the ImageJ plug-in TurboReg (www.epf.ch/thevenaz/turboreg/), StackReg (http://bigwww.epfl.ch/thevenaz/stackreg/), Matlab, or a similar. These tools support both manual registration, where the user specifies corresponding landmark points, and automatic registration methods, e.g., pyramidal least-squares minimization of the image intensities[9], maximizing mutual information[10], and normalized mutual information[11,12].

In FIB-SEM, large, pseudo-linear structures at an angle to the sectioning direction will appear to move spatially perpendicular to the sectioning direction when viewing the sections in sequence. This will misdirect typical automatic registration methods since they are unable to distinguish translations caused by drift and apparent translation caused by structures at an angle. Manual landmark annotations risk similar defects. An example of the problem can be seen in Fig. 2a, b, where a simple synthetic 3D FIB-SEM image has been generated with two spherical vesicles and a single membrane-like structure at an angle to the sectioning direction. Even though no drift is imposed here, a standard registration approach translates each image to force the membrane to be perpendicular to the image section direction,

stretching the vesicles in the process. (Fig. 2c, d), shows a similar effect on real data. The reason is that the membrane-like structure dominates the dissimilarity measure since its volume is much larger than that of the vesicles.

A problem with many registration methods is that they are agnostic to the image content. For example, real neuronal tissue contains small and large structures, and standard registration methods perform well when the angles of these structures are evenly distributed with respect to the sectioning direction. However, we have observed that this is not the case for the FIB-SEM images, we have analyzed. Firstly, for small regions of interests dominating pseudo-linear structures often appear, and at larger regions of interest, neuron processes have a tendency to be similarly oriented, resulting in inaccurate drift estimates. Thus, registration methods relying on global measures or landmark points will be less than optimal for such sections.

For improved registration, we must include models of the imaged data, such that the registration method can distinguish between drift and apparent drift caused by dominating structures at an angle to the sectioning direction. In images of neuronal tissue at nano-resolution as, e.g., the publicly available FIB-SEM dataset of the CA1 Hippocampus region of a healthy adult rodent (https://cvlab.epfl.ch/data/data-em/, $5\,nm^3$ voxel size, $1065 \times 1536 \times 2048$ voxels) we observe that vesicles are abundant, small, and on average spherical. A vesicle has a lipid bilayer shell, which physically can be modeled as an elastic material with a bending energy density functional[13,14] minimizing the curvature of the vesicle's surface. Hence, in equilibrium and without external forces, a vesicle's shape is spherical. Vesicles can take a variety of exotic shapes under special conditions[15,16], though their most probable and most common shapes are spheroids, prolates, and oblates. In this work, we propose to model the variability of the vesicle shape as ellipsoids. Since vesicles are numerous near synapses, since synapses are numerous in our images, and since vesicles on average are expected to be spherical, we can estimate the drift as the average, per section deviation from the spherical shape. We claim that this method is independent of large-scale structures such as the orientation of neuronal processes.

Our method is summarized as follows: firstly, we annotate the boundary of the vesicles by manually placing points in the images (Fig. 3a). Secondly, we fit an ellipsoid to the annotated points of each vesicle using a least-squares approach (Fig. 3b). Thirdly, we assume first that the skew of each estimated ellipsoid alone is due

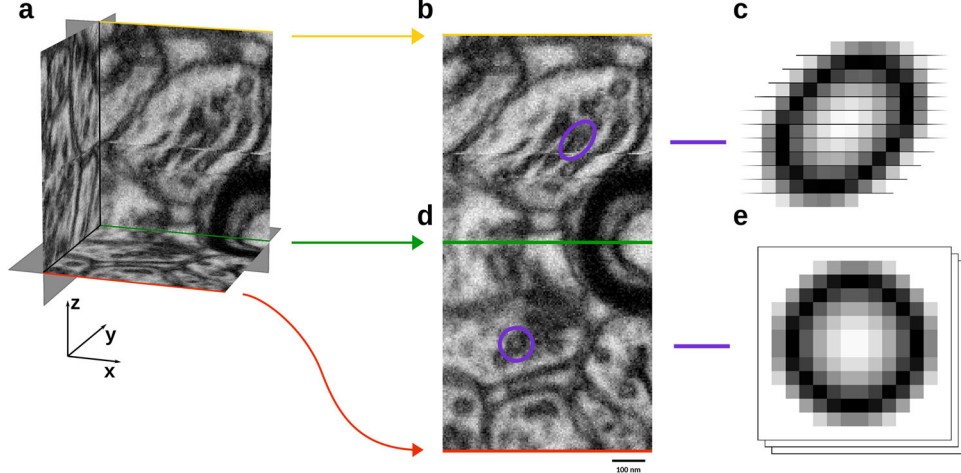

**Fig. 1 Drift introduces angles to structure in the sectioning direction, z, and is invisible in a single section. a** Small block from the publicly available FIB-SEM dataset of the CA1 Hippocampus region of a healthy adult rodent (https://cvlab.epfl.ch/data/data-em/) showing a pre-synaptic region with vesicles displaying significant drift. **b** The xz-plane in **a** highlighting the apparent effect of drift. **c** Artists depiction of a drifted vesicle as they appear in **b**. **d** The x–y plane in **a** highlighting the lack of drift effects in this plane. **e** Artists depiction of a drifted vesicle as they appear in **d**.

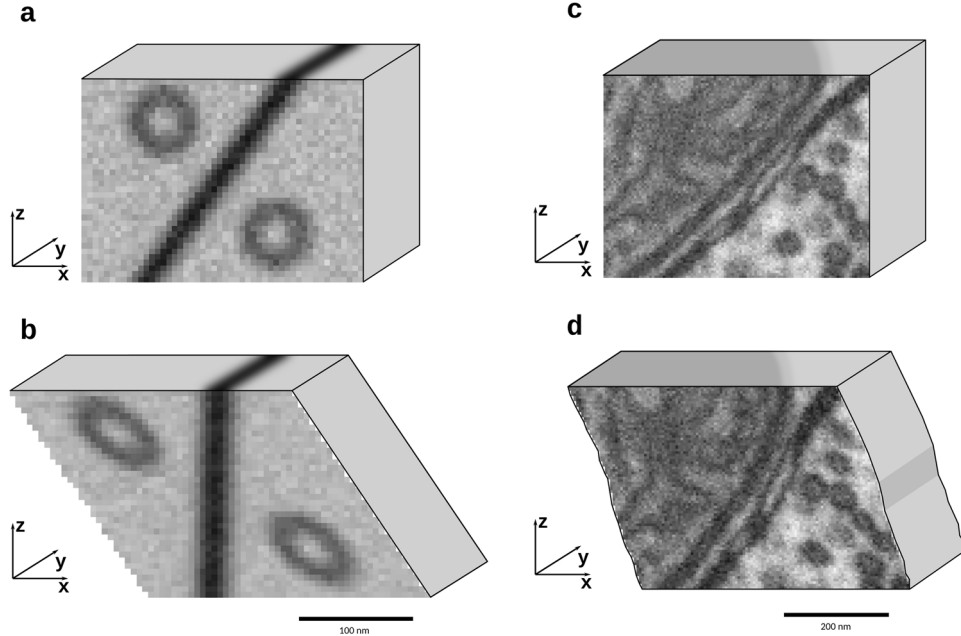

**Fig. 2 Standard registration methods confuse drift and naturally occurring trends in images. a** A synthetic image with two model perfectly spherical vesicles and a model membrane at an angle to the sectioning direction. **b** The result of using standard section-by-section registration with Mutual Information for dissimilarity measure and the implementation found in Matlab. **c** A FIB-SEM sub-volume with features which influence standard image measures adversely. **d** The result of using standard section-by-section registration displays signs of deformation by the vesicles being less spherical. This sub-volume was registered using ImageJ with StackReg and TurboReg plugins.

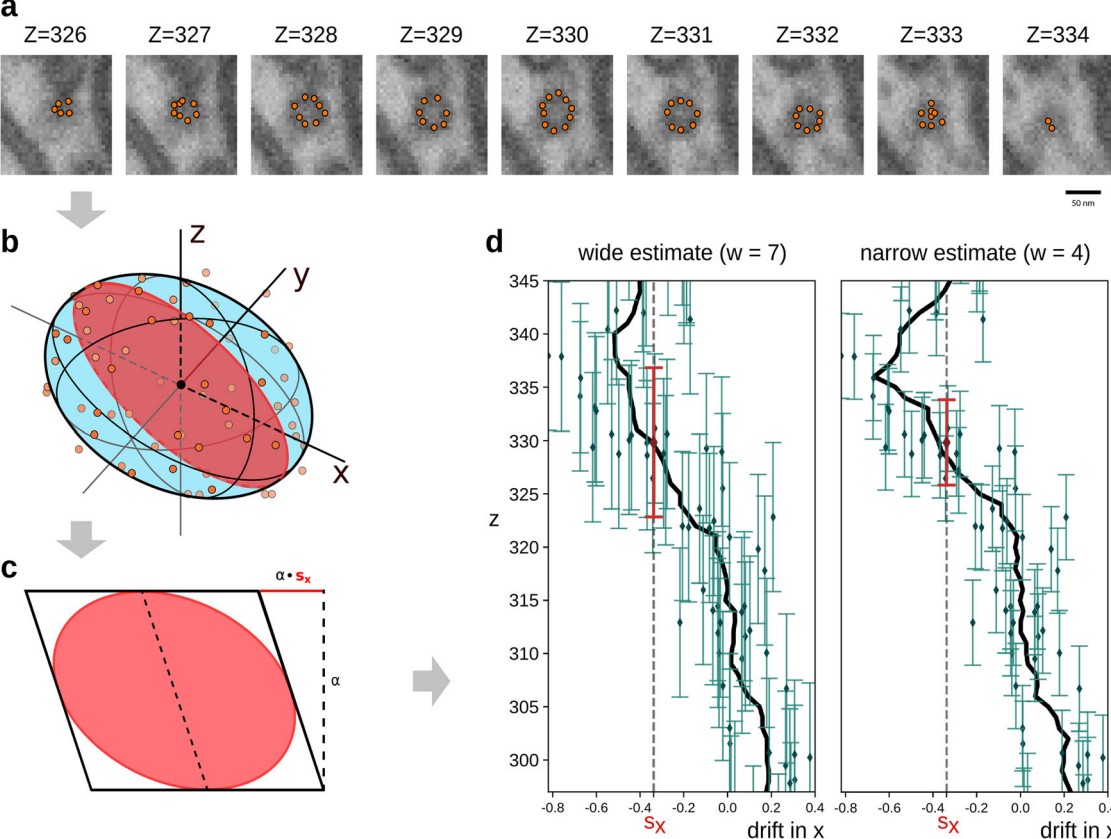

**Fig. 3 Our drift estimation uses the following four steps. a** The boundary of vesicles is manually annotated. **b** An ellipsoid (blue) is fitted to the annotated points (orange). The intersection of the fitted ellipsoid with the z–x plane is shown in **c**. **c** The skews in the zx- and yz-planes is calculated, here illustrated in the z–x plane only. **d** For each image section, the drift is estimated by the average of the drift components in x, y, and z. Diamond marker denotes a z-location of a vesicle and the bars illustrates their influence in the local average. **d** left/right shows the effect of varying the user-specified length w on the drift estimate.

to drift and calculate a drift component for each of the ellipsoids (Fig. 3c). Fourthly, we computing the average of the drift-components by regarding each drift-component estimate as a point observation at the center of the ellipsoid, and then for each section $i$, average all point observations within sections $i \pm w$ where $w$ defines a width of the estimate (Fig. 3d).

## Results

To assess the quality of drift estimates, we generated three synthetic datasets. The first two have constant drift across all sections: $(\delta x, \delta y) = (0.3 \text{ pixels}, 0.0 \text{ pixels})$ and $(0.1 \text{ pixels}, 1.0 \text{ pixels})$, and the third dataset has a drift that varies across sections (Supplementary Information and Supplementary Fig. 1). Our synthetic datasets were generated by placing random ellipsoids with uniformly random axis lengths between 3 and 6 voxels, uniformly random orientations, and uniformly random positions in a non-overlapping manner, after which the drift was added. We annotated a variety of different numbers of vesicles across image sections to assess the influence on the drift estimate. A total of 71, 97, and 283 vesicles were manually annotated in the synthetic datasets described above. Annotating the synthetic datasets took on the order of 7 h in total, which is approximately 1 vesicle per minute.

Our experiments on the synthetic datasets demonstrate that using ellipsoids to estimate the drift is significantly more accurate than standard approaches on the synthetic datasets with constant drift. Our method estimates the drift with an absolute average error of $0.22 \times 10^{-1} \pm 0.08 \times 10^{-1}$ apparently independent on the drift magnitude. In contrast, the standard registration methods estimate the drift with the error $1.12 \times 10^{-1} \pm 1.48 \times 10^{-1}$ apparently proportional to the drift magnitude. (Histograms of drift estimates are shown in Supplementary Figs. 2 and 3). The standard approaches are biased even though no larger structures like cell membrane or mitochondria are present, and we suspect that the bias is caused by subtle imbalances in the statistical distribution of image content angles in the sectioning direction.

Our synthetic image with varying drift is non-smooth. In the presence of large drift changes in the sectioning direction, the estimated drift from ellipsoids estimates are smoothed out across the change (Fig. 4). This smoothing is due to two factors: firstly, the vesicles are fitted across multiple sections, which adds some error. Secondly, the estimate is based on vesicles with the set distance, $w$ from the section. Reducing $w$ will reduce the

smoothing effect but also increase the effects of noise. Aside from the regions with large drift changes, we see a significant improvement over standard approaches (Supplementary Fig. 4). We note that the standard approaches have some variation from section to section, whereas our estimates based on ellipsoids are very stable (Supplementary Fig. 5).

Our drift estimate depends on the number of vesicles annotated, and to assess this dependence, we employ a bootstrapping approach: We use the total set of fitted ellipsoid drifts and 50000 times sample a subset of 1–200 drift-point estimates. The resulting average absolute error shows a perfect reciprocal dependence (Supplementary Fig. 6). Fitting to this bootstrapped data gives a predicted error function of $y = 0.1375x^{-0.4915}$, thus for one vesicle, an estimated error of 0.1375 is obtained, and to halve the error, your roughly need to annotate four times the vesicles. For the real FIB-SEM dataset, we manually annotated 961 vesicles. This dataset has 1065 pixels in the sectioning direction distanced 5 nm apart. Assuming that the average height of a vesicle is 45 nm this implies that there a vesicle on average is seen in nine sections. Thus, on average we have annotated $961 \cdot 9/1065 \cong 8.12$ vesicles per section, and the estimated error is $0.1375 \cdot 8.12^{-0.4915} \cong 0.049$ pixels $\cong 0.25$ nm for each section.

The drift estimation accuracy also depends on the variation in radii of the vesicles. Specifically, in the extreme case that the vesicles are exactly spherical in the without drift, we only need a single vesicle to estimate a constant drift exactly since the drift-component will be equal to the drift. We therefore also assessed the correspondence between both radii variation and drift magnitude on the estimation error. We generate a fourth synthetic image with 50000 vesicles for a variety of drift magnitudes and vesicle radii and measure the mean absolute error using ten vesicles for each estimate (Supplementary Fig. 7). Firstly, the figure shows a clear dependence on the variation of the radius. If radius does not vary, and if each vesicle is a sphere, then there is no error, but the error increases as the variation in radius increases. Secondly, the figure shows that the estimate is unaffected by the drift magnitude. Thus, estimating a sub-pixel drift and large drift will give an error of equal size.

For this work, we compare our method with the registration methods using the sum of squared difference (SSD) and mutual information (MI), normalized mutual information (NMI), and normalized cross-correlation (NCC) as dissimilarity measures. For completeness, we also experimented with correction using phase correlation[17] and compared with built-in registration

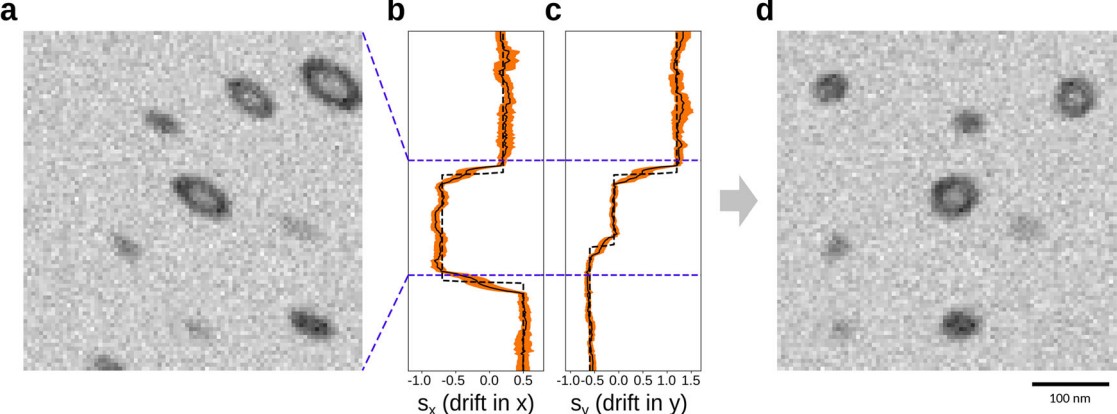

**Fig. 4 On synthetic images, our local drift estimates are precise for when the drift is constant and show smooth transition when drift varies. a** A small subsection of a drifted synthetic image shown in a side view (zx-plane). **b, c** The estimated ellipsoid drift in x and y, respectively, as a function of z. Stippled curve is the ground truth, black curve is the estimate, and orange shows the variation in the per-vesicle estimates. **d** The same synthetic image corrected in z–x plane.

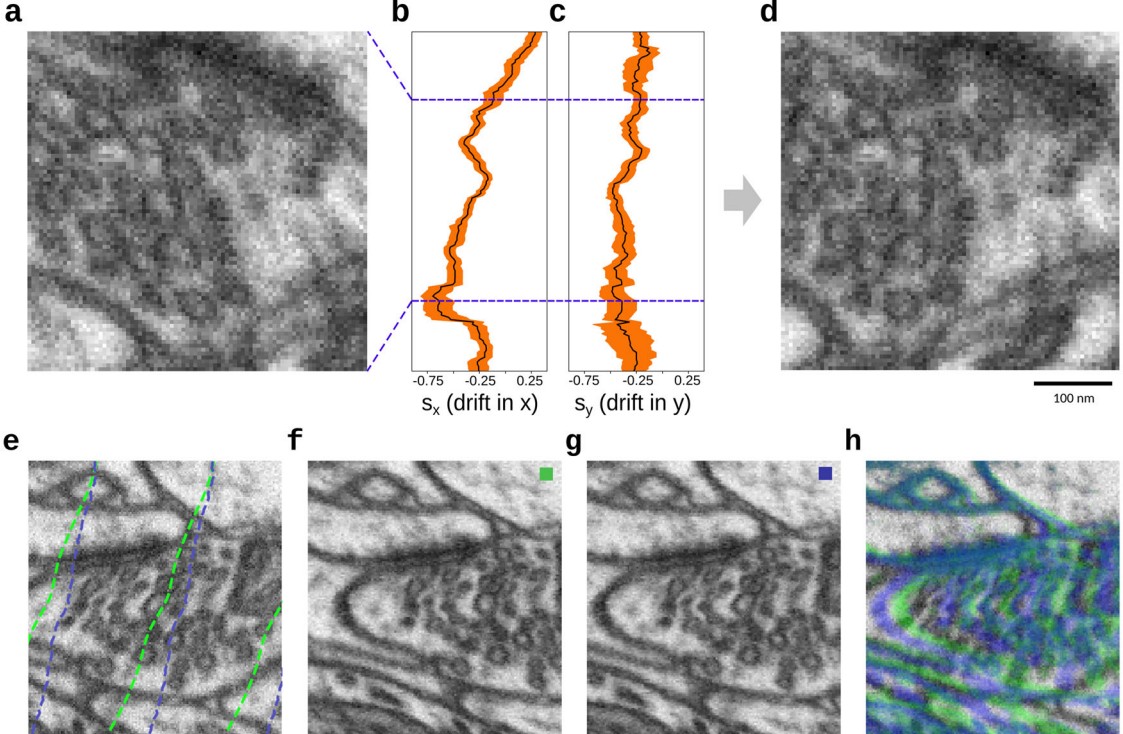

**Fig. 5 Our drift estimates show similar behavior on real data as on the synthetic data. a** A small subsection of a drifted FIB-SEM image shown in a side view ($z$–$x$ plane). **b** The estimated ellipsoid drift in $x$ as a function of $z$. **b**, **c** The estimated ellipsoid drift in $x$ and $y$, respectively, as a function of $z$. Black curve is the estimate, and orange shows the variation in the per-vesicle estimates. **d** The same FIB-SEM image corrected in $z$–$x$ plane. **e** Comparison of drift estimated by our proposed method (green) and using normalized mutual information image measure (blue). **f** The resulting registration using our proposed method. **g** The resulting registration using normalized mutual information. **h** Pseudo-color overlay of registration results obtained by our proposed method compared to registration using normalized mutual information.

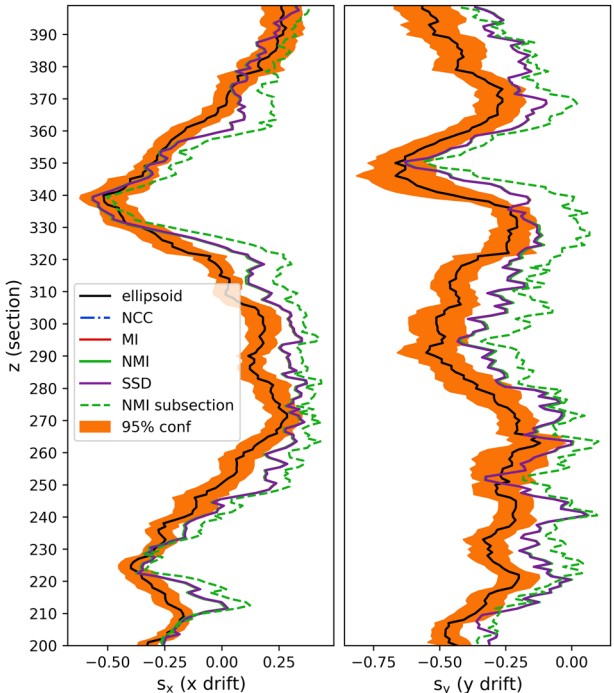

**Fig. 6 Drift estimates for standard registration method are consistent and most often outside the 95% confidence interval of our method.** (Left) and (right) show the estimates in the $x$- and $y$-direction. The stippled curve shows the result of applying the standard registration method on a subsection of the image, which is dominated by a large process.

implementations in Matlab. Not reported here, we also experimented with optical flow estimation as implemented in OpenCV across the pairwise image section using only subregions of the images with vesicles. Our method is clearly superior to the state-of-the-art global registration methods on the synthetic images when compared to the ground-truth drift.

For real FIB-SEM images, the ground-truth drift values usually do not exist. Hence, the following conclusions are based on our experience with synthetic data described above. In the publicly available FIB-SEM dataset of the CA1 Hippocampus region of a healthy adult rodent (https://cvlab.epfl.ch/data/data-em/), we observe a significant non-zero drift signal (Fig. 5). After drift-correction using our drift-estimates, the vesicles look visually less ellipsoidal. Qualitatively comparing the drift estimate on the real with the synthetic images, we find that the estimate on the real dataset is similarly distinctly different from standard approaches (Fig. 6 and Supplementary Fig. 8). The estimates by the individual standard approaches can be seen to be in close agreement with each other with regards to both the magnitude and direction of the translation. However, for subregion in the real dataset, we also see that standard methods are biased with respect to image content (Fig. 6). Asserting the section-wise drift, we observe rapid changes in drift estimates similarly to the synthetic image with varying drift (Supplementary Fig. 9). Hence, we expect some estimation error and smoothing effects to be present for the real data as well.

## Discussion

To conclude, FIB-SEM images often suffer from sub-pixel drift often in the order of 0.5 pixels, and this drift accumulates across several slices resulting in distortion of distance and shape

measures in 3D. Standard registration methods fail to correct this drift, as these methods cannot distinguish between drift and naturally occurring slopes in the sectioning direction. We have discovered that due to the abundance of vesicles in neuronal tissue and their biomechanical properties, they function well as statistical markers for drift, and we have presented a simple method for identifying and correcting the drift. We have compared our method with state-of-the-art registration methods based on global measures, and our method has proven to be more accurate. We encourage correction of drift in neuronal tissue whenever the analysis is of or relies on the geometry of the structures. Further work should be carried out to assess the effect of the drift on biological images and to develop less labor-intensive methods for estimating and correcting this drift.

## Methods

Our method consists of the following sequence of steps: (1) Annotating vesicle boundary using points. (2) obtaining ellipsoids by least squares estimation on the boundary points. (3) calculate drift parameters for each ellipsoid. (4) Estimate the average drift locally to each section using the ellipsoids drift parameters in conjunction with their position. Our method is implemented and available online[18].

**Choice of ellipsoid representation**. Let $x$, $y$, $z$ be the axes of an image with $x$, $y$ the plane of each image section and $z$ the image sectioning axis. An ellipsoid centered at the origin can be described implicitly by the quadratic surface equation $\mathbf{u}^T H \mathbf{u} = 1$, where $\mathbf{u} = [x, y, z]^T$, and $H$ is a $3 \times 3$ symmetric positive definite matrix,

$$H = \begin{bmatrix} A & D & E \\ D & B & F \\ E & F & C \end{bmatrix}. \tag{1}$$

We will refer to the elements of $H$ as the parameters of the ellipsoid.

**Obtaining ellipsoids from boundary points**. We fit an ellipsoid to the vesicle membrane points of each vesicle in 3D. At least 9 non-degenerate points are required for a unique fit of an ellipsoid with an arbitrary center, radii, and rotation. We choose to fit the ellipsoids given as $\mathbf{u}^T H \mathbf{u} = 1$ using a least-squares approach[19] which we implemented in Python. We briefly compared this method to a numeric gradient decent approach minimizing the squared distance in the Euclidean norm as it has a slight difference in result compared to the algebraic norm defined by the ellipsoid equation. Our results showed that the algebraic norm minimization method produced slightly more accurate drift estimates while being many orders of magnitude faster than the numeric decent method.

**Estimating drift from ellipsoid parameters**. We define the drift in the image as a sideways translation of each image section with respect to the previous section. Let $\delta x$, $\delta y$ be the amount of translation by which some section is translated with respect to the previous section and let $\Delta z$ denote the distance between subsequent sections. We represent the translation as a shear map with shear coefficients $s_x = \delta x / \Delta z$, $s_y = \delta y / \Delta z$. If we assume the drift is constant as a function of $z$, we can represent the drift as one single mapping $S$ given by

$$S\mathbf{x} = \begin{bmatrix} 1 & 0 & s_x \\ 0 & 1 & s_y \\ 0 & 0 & 1 \end{bmatrix} \begin{bmatrix} x \\ y \\ z \end{bmatrix} = \begin{bmatrix} x + s_x z \\ y + s_y z \\ z \end{bmatrix} = \mathbf{u}. \tag{2}$$

The shear mapping is a non-singular linear transformation, and since $S^{-1}S$ is the identity transformation, the quadratic equation is still solved when

$$1 = \mathbf{x}^T H \mathbf{x} = \mathbf{x}^T \left(S^{-1}S\right)^T H S^{-1} S \mathbf{x} = \mathbf{u}^T S^{-T} H S^{-1} \mathbf{u}. \tag{3}$$

Thus, transforming each point on the ellipsoid by $S$ corresponds to a new quadratic surface defined by the matrix representation $\hat{H} = S^{-T} H S^{-1}$, or equivalently $H = S^T \hat{H} S$. Since an ellipsoid is a quadratic surface with a closed surface, and since non-singular linear transformations on closed surfaces cannot produce open surfaces, we conclude that the result is still a closed surface defined by a quadratic surface, i.e., an ellipsoid, spheroid or sphere.

Let $\hat{H}$ be the shear-transformed ellipsoid estimated from data with elements $\hat{A}$, $\hat{B}$, $\hat{C}$, $\hat{D}$, $\hat{E}$, and $\hat{F}$. The values of $\hat{E}$ and $\hat{F}$ gives the shape of the ellipsoid in the $z$–$x$ and the $y$–$z$ plane, i.e., the tilt of the ellipsoid as a function of $z$. An untilted ellipsoid is symmetric across the plane $z$ and has $E = F = 0$. Defining the untilted ellipsoid in terms of a shear-transformed tilted ellipsoid $H = S^T \hat{H} S$, we set $E = F = 0$ and solve for $s_x$ and $s_y$. We get

$$s_x = \frac{\hat{D}\hat{F} - \hat{B}\hat{E}}{\hat{A}\hat{B} - \hat{D}^2}, \, s_y = \frac{\hat{D}\hat{E} - \hat{A}\hat{F}}{\hat{A}\hat{B} - \hat{D}^2}. \tag{4}$$

Let $\mathbf{s} = (s_x, s_y)^T$ represent the shear of some ellipsoid. By assumption, each ellipsoid is rotated uniformly at random. Thus, it follows that given no drift in the

data, we should have $\mathbb{E}[\mathbf{s}] = \mathbf{0}$, since by an argument of symmetry, a tilt in any direction should be equally likely. Assume now we add some drift $\mathbf{k}$, giving rise to new shear parameters $\mathbf{s}'$. Since the composition of shear transformations simply amounts to adding the shear parameters, and by the linearity of expectation, we get

$$\mathbb{E}[\mathbf{s}'] = \mathbb{E}[\mathbf{s} + \mathbf{k}] = \mathbb{E}[\mathbf{s}] + \mathbb{E}[\mathbf{k}] = \mathbf{k}. \tag{5}$$

Thus, given $N$ fitted ellipsoids with $\mathbf{s}^i$ the vector of shear constants for ellipsoid $E_i$, $1 \le i \le N$, we estimate the drift in the images $\mathbf{k}$ by the average drift,

$$\mathbf{k} = \frac{1}{N} \sum_{i=1}^{N} \mathbf{s}'^{(i)} \tag{6}$$

Enumerating the image sections by $I_j$, $1 \le j \le M$ such that $I_1$, ..., $I_N$ are ordered with increasing $z$ choosing $I_1$ as the reference image, drift correction can be obtained by transforming $I_j$ by $S^{-(j-1)}$.

**Drift correction assuming varying drift**. Since drift in images may vary, e.g., due to manual correction during the scanning operation, movement of the sample, or charge equalization, it is likely that the amount of drift varies in the sectioning direction. Given a large enough population of ellipsoids, it is possible to estimate the drift per image section. To accomplish this, we view the individual estimated ellipsoid-drifts as a point-estimate at the center of the ellipsoid. Specifying a width $w$ of the point estimate, we compose a drift estimate for each section by the average of the ellipsoids with center closer than $w$ to the section. Denoting $d(E_i, j)$ the perpendicular distance in the section direction from ellipsoid $E_i$ to section $j$, we can write the drift estimate $\mathbf{k}_j$ of section $j$ as

$$\mathbf{k}_j = \sum_{i=1}^{N} \frac{1_{d(E_i, j) < w}}{\sum_{n=1}^{N} 1_{d(E_n, j) < w}} \mathbf{s}'^{(i)}, \tag{7}$$

where $1_P$ takes the value 1 when P is true and 0 otherwise. In the absence of visible vesicles in one or multiple sections, we suggest either interpolating the drift parameters from nearby known values or assume the drift is zero, depending on the dataset.

**Reporting summary**. Further information on research design is available in the Nature Research Reporting Summary linked to this article.

## Data availability

Our synthetic, derived data, and data for producing our figures can be provided upon request. In this paper, we have discussed the publicly available fib-sem data from https://cvlab.epfl.ch/data/data-em/.

## Code availability

The code for producing our figures can be provided upon request. The drift correction application is available at (https://doi.org/10.17894/ucph.b61d5ca9-53df-4909-92ee-f8ee026e39bb). The accompanying source code is available upon request[19].

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

## Acknowledgements

This work was funded by the Villum Foundation through the Center for Stochastic Geometry and Advanced Bioimaging (CSGB). We would like to thank Stine Hasselholt from the Department of Clinical Medicine at the University of Aarhus in particular for her biological insights, which helped us greatly in conceptualizing the method.

## Author contributions

H.J.T.S. developed the method, performed experiments, and wrote the paper in collaboration with and under supervision of J.S. S.D. provided state-of-the-art comparison registrations.

## Competing interests

The authors declare no competing interests.
