## [Peer Review File · Communications Biology]

Reviewers' comments:

Reviewer #1 (Remarks to the Author):

The paper discusses a method of drift correction in 3D electron scanning microscopes, in neuronal tissue images. The authors show an example, where the standard methods fail to correct the drift properly, and propose an alternative method, based on the analysis of specific image structures, that are annotated manually, which outperforms general methods in specific cases.

The proper reconstruction of 3D electron microscope image is surely an important processing step, and therefore the paper of interest to many researchers. The text is well organized and easy to comprehend. However, it suffers from various omissions and understatements that need to be clarified, before it merits for publication in Communications Biology.

Specific comments:

- Line 8- Abstract: 'in the order of 0.5 pixels'- authors should provide the physical (in nm) order of the drift, not in pixels. If we perform the scanning with a different resolution- e.g. the pixels size is 10 nm instead of 5 nm, will then the drift increase by the factor of 2? By providing the drift in pixels, the authors, improperly, suggest that the drift is related to the scanning resolution. Moreover (in the Introduction), the authors should clarify where the magnitude '0.5 pixels' comes from- their own observations, average of different values in the literature?

- Line 18, 21- authors refer to FIB-SEM as a method that introduces the aforementioned drift.

- However, that drift is an imaging modality present in other serialsectioning scanning techniques, e.g. 3View.

- In the Introduction the authors should be more specific what they mean by the drift: translations, translations+dilation, etc...? They specify it in Methods, line 216, 'We define the drift in the image as a sideways translation of each image section with respect to the previous section' - this section should be moved to the Introduction.

- The authors do not provide anywhere the pixel/voxel size, nor the total sample dimensions

- Figures 1-5- there are no scale bars present.

- Figure 2- The authors show the failure of the standard alignment methods, using synthetic model image. Could possibly the authors show as well such an improper alignment using real data, to support the claims made in the Introduction?

- The authors provide the detailed comparison of the proposed method performance, to the performance of several standard alignment method. The authors use the simulated data, where the ground truth is known, for such a comparison. However, the comparison using real data is absent. Contrary to the case of the simulated data, the ground truth cannot be known, however, at least the qualitative comparison is available, for example an overlay of two color channel- one the aligned sections with the proposed method, the other one with the reference method (e.g. additional panels in Fig. 5: e) real data before drift correction, f) real data after drift correction with the proposed method, g) real data after drift correction with the reference method, h) pseudocolor overlay. In this way, a reader could possibly see an actual example of the proposed method superiority.

- Equation 2- unnecessary 0 in the first matrix (top right position).

- Supplementary Figures 4 & 5- an overlay of several different colors is quite messy, it would be more clear to present the results as box plot bars.

Reviewer #2 (Remarks to the Author):

This manuscript presents a restoring method for drifted FIB-SEM volume image with assumption of vesicle shapes.

Frankly, this method is only effective for FIB-SEM volume images where only spherical vesicles were observed. To extend general volume images that include other bio-components with spherical vesicles, I considered that it was necessary to include a method for automatically determining (segmenting) the vesicles. According to the Method section, the authors annotated vesicle boundary. However, detailed results for Figure 5 were not presented. Therefore, vesicles that have changed shape are indistinguishable from other vesicles. For these, I think we need to add results and discussions.

In addition, it is desirable to add samples for different biological observations in order to publish in a journal for biology.

I believe that this manuscript should be revised majorly.

The comments were given here.

1. I considered that the title of this manuscript was too exaggerated. At the very least, it should be clear that this manuscript was restoring with assuming the shape of a vesicle. For example, "with assumption of vesicle shapes" can be inserted in the title.
2. Regarding Figure 5, annotated vesicles should be marked to distinguish the other vesicles. The authors should discuss about performance-relations between drift-restoring and vesicle-detections.
3. It is preferable to present restored images (corresponding to Suppl.-Fig. 3) by several methods in SI.

Reviewer Comment Response

Dear editor,

In this document, we discuss all comments from the reviewers. It is organized in a per-reviewer order, where each comment has been enumerated, with the reviewer's comment followed by our response.

On behalf of the authors, Jon Sparring

Reviewer 1 Comments

Overview

The paper discusses a method of drift correction in 3D electron scanning microscopes, in neuronal tissue images. The authors show an example, where the standard methods fail to correct the drift properly, and propose an alternative method, based on the analysis of specific image structures, that are annotated manually, which outperforms general methods in specific cases.

Overall verdict

The proper reconstruction of 3D electron microscope image is surely an important processing step, and therefore the paper of interest to many researchers. The text is well organized and easy to comprehend. However, it suffers from various omissions and understatements that need to be clarified, before it merits for publication in Communications Biology.

Response: Thank you for your suggestions.

Comment 1.1

Line 8- Abstract: 'in the order of 0.5 pixels'- authors should provide the physical (in nm) order of the drift, not in pixels. If we perform the scanning with a different resolution- e.g. the pixels size is 10 nm instead of 5 nm, will then the drift increase by the factor of 2? By providing the drift in pixels, the authors, improperly, suggest that the drift is related to the scanning resolution.

Response: We agree with the reviewer and have changed the document such that both physical and pixel units are given. (see line 8)

Comment 1.2

Moreover (in the Introduction), the authors should clarify where the magnitude '0.5 pixels' comes from- their own observations, average of different values in the literature?

Response: This is our own observation on the dataset examine. We have clarified this in the text. (see line 8)

Comment 1.3

Line 18, 21- authors refer to FIB-SEM as a method that introduces the aforementioned drift. However, that drift is an imaging modality present in other serial sectioning scanning techniques, e.g. 3View.

Response: This is correct, and therefore, we have listed other modalities, where the drift has been observed. (see lines 20-21).

Comment 1.4

In the Introduction the authors should be more specific what they mean by the drift: translations, translations+dilation, etc...? They specify it in Methods, line 216, 'We define the drift in the image as a sideways translation of each image section with respect to the previous section' - this section should be moved to the Introduction.

Response: That is a good idea, we now include the definition in the introduction as well. We have added that we mean “sideways” translation in the Introduction to mirror the definition in the Methods section. (see lines 23-25).

Comment 1.5

The authors do not provide anywhere the pixel/voxel size, nor the total sample dimensions

Response: Thank you for your comment, we have added the information. (see line 85).

Comment 1.6

Figures 1-5- there are no scale bars present.

Response: Thank you. We have added scale bars were relevant. (see Figures 1-5).

Comment 1.7

Figure 2- The authors show the failure of the standard alignment methods, using synthetic model image. Could possibly the authors show as well such an improper alignment using real data, to support the claims made in the Introduction?

Response: Thank you. We have found and displayed a region showing similar problems in real data. (See updated Figure 2 and accompanying caption lines 67-69).

Comment 1.8

The authors provide the detailed comparison of the proposed method performance, to the performance of several standard alignment method. The authors use the simulated data, where the ground truth is known, for such a comparison. However, the comparison using real data is absent. Contrary to the case of the simulated data, the ground truth cannot be known, however, at least the qualitative comparison is available, for example an overlay of two color channel- one the aligned sections with the proposed method, the other one with the reference method (e.g. additional panels in Fig. 5: e) real data before drift correction, f) real data after drift correction with the proposed method, g) real data after drift correction with the reference method, h) pseudocolor overlay.

In this way, a reader could possibly see an actual example of the proposed method superiority.

Response: Thank you for the suggestion. We think this is a good way to compare the methods in a qualitative manor. We have added the suggested panels. (See the updated Figure 5).

Comment 1.9

Equation 2- unnecessary 0 in the first matrix (top right position).

Response: The reviewer is correct. The extra 0 has been removed in revised manuscript. See (1)

Comment 1.10

Supplementary Figures 4 & 5- an overlay of several different colors is quite messy, it would be more clear to present the results as box plot bars.

Response: We agree and have changed the visualization to be box-plots instead. (see Supplementary Figures 4 & 5 also with slight modifications to the captions).

Reviewer 2 Comments

Overview

This manuscript presents a restoring method for drifted FIB-SEM volume image with assumption of vesicle shapes.

Overall verdict

Comment 2.1

Frankly, this method is only effective for FIB-SEM volume images where only spherical vesicles were observed.

Response: It is correct that in this paper, we consider spherical vesicles, that is, vesicles which on average are spherical. It is certainly an interesting project to investigate other simple geometries for sub-pixel accuracy in the registration method. Initial investigations have shown that, assuming random rotations, similar properties are obtained for less symmetric objects, however, it is beyond the scope of this work to generalize to other shapes.

Comment 2.2

To extend general volume images that include other bio-components with spherical vesicles, I considered that it was necessary to include a method for automatically determining (segmenting) the vesicles.

Response: With this paper we wish to focus on the limitation of state-of-the-art registration methods, which is, that they are agnostic to the meaning of the image content. For this, an automatic method is not needed.

Comment 2.3

According to the Method section, the authors annotated vesicle boundary. However, detailed results for Figure 5 were not presented. Therefore, vesicles that have changed shape are indistinguishable from other vesicles. For these, I think we need to add results and discussions.

Response: Perhaps we have been unclear. We are performing a global transformation on each section. We annotate some vesicles, but all vesicles change shape, since we translate each image section accordingly. Therefore, in Figure 5, all vesicles have changed shape such that the average shape near a given section is spherical. Vesicles used for correction is not necessarily visible in the shown image. To illustrate the effect of the global transformation, we have updated Figure 2 to demonstrate the global effect on a real example, and we have clarified Figure 5 with added panels.

Comment 2.4

In addition, it is desirable to add samples for different biological observations in order to publish in a journal for biology.

Response: Our focus is on a common methodology used in the biology domain: registration. A limit of state-of-the-art registration methods is that they do not model what is being imaged. We explain and demonstrate this limitation, and our paper demonstrates a method to improve the results. Although we do not examine a particular biological problem, we find it an important result particularly for the

audience of biology journals. However, drift is observed in other modalities, and in line 19, we have included an extended list.

Comment 2.5

I believe that this manuscript should be revised majorly.

Response: Thank you for your comments, we have considered every point carefully and made appropriate improvements.

Comment 2.6

I considered that the title of this manuscript was too exaggerated. At the very least, it should be clear that this manuscript was restoring with assuming the shape of a vesicle. For example, “with assumption of vesicle shapes” can be inserted in the title.

Response: We agree that the title can be more specific, since we rely on statistical and size properties of objects, and thus, vesicles are a prime target. We have therefore changed the title to "Restoring drifted **electron microscope** volumes **using synaptic vesicles** at sub-pixel accuracy"

Comment 2.7

Regarding Figure 5, annotated vesicles should be marked to distinguish the other vesicles.

Response: Thank you, since the vesicles are randomly selected by our algorithm, there is no annotated vesicle in a given restored subvolume of the size shown in Figure 5. However, the effect of the annotated vesicles determines the estimated drift throughout the entire section. As discussed in response to Comment 2.8, we have on average annotated about 8 vesicles per section in the real data set. The improved visualization of this effect is given in Figure 5e-f.

Comment 2.8

The authors should discuss about performance-relations between drift-restoring and vesicle-detections.

Response: Thank you. We have already reported the total time for annotating 451 vesicles (line 119) and the relation between the number of vesicles and accuracy of our drift correction method (Supplementary Figure 8). We have added a comment on our approximate annotation speed (line 119) and a more precise discussion on the relation between the error and the number of vesicles annotated, see caption for Supplementary Figure 8 and accompanying text lines 143-149.

Comment 2.9

It is preferable to present restored images (corresponding to Suppl.-Fig. 3) by several methods in SI.

Response: We agree that it is important to visually confirm the difference in effect of standard methods with our methods. Thus, we have included new panels, see Figure 5e-h.

REVIEWERS' COMMENTS:

Reviewer #1 (Remarks to the Author):

The authors improved significantly the manuscript in the revision, and provided detailed responses to all specific comments. I suggest publishing the manuscript in the current form.